# Let's (not) get together! The role of social norms on social distancing during COVID-19

**Déborah Martínez[1], Cristina Parilli[2], Carlos Scartascini**[ORCID]**[1]\*, Alberto Simpser[3]**

**1** Research Department, Inter-American Development Bank, Washington, DC, United States of America, **2** IDB Invest, Inter-American Development Bank, Washington, DC, United States of America, **3** Political Science Department, ITAM, Mexico City, CDMX, Mexico

\* carlossc@iadb.org

**Data Availability Statement:** All relevant data are within the manuscript and its Supporting information files.

**Funding:** The Inter-American Development Bank and ITAM provided support in the form of salaries

## Abstract

While effective preventive measures against COVID-19 are now widely known, many individuals fail to adopt them. This article provides experimental evidence about one potentially important driver of compliance with social distancing: social norms. We asked each of 23,000 survey respondents in Mexico to predict how a fictional person would behave when faced with the choice about whether or not to attend a friend's birthday gathering. Every respondent was randomly assigned to one of four social norms conditions. Expecting that other people would attend the gathering and/or believing that other people approved of attending the gathering both increased the predicted probability that the fictional character would attend the gathering by 25%, in comparison with a scenario where other people were not expected to attend nor to approve of attending. Our results speak to the potential effects of communication campaigns and media coverage of compliance with, and normative views about, COVID-19 preventive measures. They also suggest that policies aimed at modifying social norms or making existing ones salient could impact compliance.

## Introduction

Since the COVID-19 pandemic began in early 2020, much has been learned about how infection can be prevented. In particular, social distancing and avoiding indoor gatherings have emerged as some of the most powerful and effective preventive behaviors [1]. Despite the strength of the evidence on the dangers of close social contact [2, 3], many people continue to gather with friends and to participate in social events [4–6], which has helped the virus to potentially spread even to the highest political circles [7, 8]. If the pandemic is to be contained, it is crucial to understand what drives people to engage in behavior that is inconsistent with the available scientific evidence and public health guidelines [9].

The problem does not appear to be one of information or credibility, as survey evidence shows that most people agree that social gatherings ought to be avoided. As far back as May of 2020, 79.5% of survey respondents in the United States agreed that gatherings of 10 or more people should not be allowed [10]. In Mexico, the country where we conducted the present study, 82% of those surveyed in April of 2020 approved of the public health guidelines in place,

for all the authors, but did not have any additional role in the study design, data collection and analysis, decision to publish, or preparation of the manuscript. The specific roles of these authors are articulated in the 'author contributions' section. The authors affiliations does not alter our adherence to PLOS ONE policies on sharing data and materials as it is articulated in the 'Methods' and 'Additional information' sections.

**Competing interests:** The authors declare no competing interests. The Inter-American Development Bank and ITAM provided support in the form of salaries for all the authors, but did not have any additional role in the study design, data collection and analysis, decision to publish, or preparation of the manuscript. The specific roles of these authors are articulated in the 'author contributions' section. The authors affiliations does not alter our adherence to PLOS ONE policies on sharing data and materials as it is articulated in the 'Methods' and 'Additional information' sections.

which included restrictions on mass gatherings [11]. According to our own data, 73% of people recognize that gathering in enclosed spaces, such as restaurants, represents a high risk for contracting COVID-19. Still, about 43% recognize having visited friends and family in their homes during the previous week.

In this article, we investigate the role of *social norms* on compliance with preventive behaviors—specifically with social distancing. We do so by conducting a survey experiment on more than 23,000 individuals in Mexico. The experiment consists of a vignette, described in the form of a story, depicting a fictional individual, *Mariana*, who has been invited to attend a friend's birthday gathering and must decide whether or not to attend. This story portrays a situation that most Mexicans can relate to (birthday celebrations) and what the literature highlights to be individuals' relevant reference network during the current pandemic (family and friends) [12]. These social gatherings are also relevant because they have been shown to lead to super-spreading events [2, 3]. The treatments randomly assign respondents to different social norms prompts, providing information on Mariana's beliefs about: (i) whether other invitees *will* attend the gathering (empirical expectations), and (ii) whether other invitees *approve of* others' attending the gathering (normative expectations). After being exposed to the social norms prompt, respondents are asked to state whether they believe that Mariana will attend the gathering, and whether they believe that Mariana should attend the gathering.

We find that the prompt about whether others are likely to attend has a strong effect on the respondent's prediction as to whether Mariana will attend the gathering or not. These findings are in line with prior findings, in settings other than the current COVID-19 pandemic, that individuals tend to conform to what they perceive is the prevailing behavior [13–18]. Interestingly, we find no effect of any of the treatments on respondent predictions about what Mariana *ought to* do: the overwhelming majority believe she should not attend.

## Theoretical background

It has long been argued that individual behavior is strongly influenced by what others do (*descriptive norms*) and what others approve doing (*prescriptive or injunctive norms*) [14, 16, 19–21]. The literature accords different roles and effects to descriptive versus injunctive norms [22]. Descriptive norms indicate those cases in which you prefer to carry out an activity because you believe it meets your needs (unconditional preference) or because you expect others to do it (conditional preference). Injunctive norms indicate those cases in which you prefer to engage in an activity because you believe it is the right thing to do (unconditional preference), or because you expect others to engage in the activity and believe that others think that you should do so as well (conditional preference). In this latter case of conditional preferences, choices and behaviors depend on both empirical expectations (what you believe others are doing) and normative expectations (what you believe others think you should do) [22].

In our setup, a social norm is a rule that maps empirical and normative expectations onto behaviors. A social norm is followed by individuals in a population *"on the condition that they believe that (i) most people in their reference network conform to it (empirical expectation) and (ii) that most people in their reference network believe they ought to conform to it (normative expectation)"* [22, p.5].

Both empirical and normative expectations have been shown to influence behavior. Policymakers, for example, have increasingly made use of social norms to nudge individuals in diverse contexts, with goals such as reducing medical prescriptions, increasing tax compliance, and reducing energy and water consumption [23–28], and social norms can also affect willingness to enforce and sanction violations [29–31].

Social norms could be extremely relevant for explaining and affecting behaviors during the current pandemic [9, 32]. Goldberg et al. [12] and Smith et al. [33] find that an individual's perceptions about how many others abide by social distancing correlate with the individual's propensity to social distance herself, and the effect of social norms can be stronger on individuals lacking a sense of duty [34]. As people seek to conform or to imitate the behavior of others [13], news coverage of celebrities or political leaders failing to abide by, or criticizing, preventive behaviors [35, 36] could in fact reduce public compliance with such behaviors, as they might be "normalizing" them in the eye of the public [37–39]. However, norm-based interventions and media coverage on events showing compliance with preventive behaviors can potentially help [40]. Still, it is worth noting that norm-based messages might not have any differential effect on the understating of COVID-19 guidelines [41] and that norm nudges need to include more than informative messages to be effective [42]. These findings make it even more important to investigate how and why social norms would change people's compliance with preventive behaviors in order to further refine future interventions and massive communication efforts.

Bicchieri et al. [43] run a survey experiment similar to ours where normative and empirical expectations are randomly varied in a 2-by-2 schema, and respondents are then asked to predict the compliance of a fictional third party with social distancing. That study, like ours, finds that assignment to the condition with "high" normative and empirical expectations promoted compliance. However, our approaches differ in three important dimensions. First, instead of asking whether the third party would abide by *social distancing in general*, we confront the respondent with a *very specific scenario*: whether or not to attend the birthday party of a close friend. We believe that our approach is more vivid and therefore less prone to eliciting abstract responses colored by social desirability biases or demand effects. Second, instead of using a Likert scale we force a dichotomic yes/no response that mimics many social distancing choices: one can either attend a gathering or refrain from attending. Third, we elicit both predicted behavior and respondent normative views, which allows us to study whether any effects on (predicted) behavior might be underpinned by, or correlated with, effects on normative assessments.

Our paper builds on a recent but strong behavioral literature studying behaviors associated with the current COVID-19 pandemic that attempts to promote preventive behaviors and a more effective pandemic response [9]. Caparo & Barceló [44] show that individuals primed with "reasoning" messages are more willing to wear facemasks than those primed to "rely on their emotions", which points out that people's compliance can be increased if they are not driven by emotions in their decision-making. Highlighting the risks associated with not following social distance have a larger effect than providing information [45]. "Deontological" messages, based on people's duty to do the right thing for their families and friends, seems to be more effective than utilitarian or moral messaging [46]. Along this line, [47, 48], and [49] findings are also consistent with the idea that prosocial motivation is effective in promoting intention to comply with preventive behaviors, particularly if they are able to develop individuals' empathy towards those more vulnerable to being infected [50].

These findings are relevant as they allow us to understand how individuals perceive and act according to the consequences of their own personal actions on others. Thus, this lays the groundwork to go even further and also understand how individuals react when faced with the behavior of others –that is, how perceived social norms can change individuals' behavior even if they were personally willing to comply with preventive measures due to prosocial motives. Can the perception of what others do and approve of change individuals' intentions of complying with public health guidelines? Our study aims to contribute to the related literature and complement other similar studies conducted during the pandemic.

## Methods

### Participants

Our survey experiment was part of a broader COVID-19-focused survey in Mexico, approved by the IRB of the Instituto Tecnológico Autónomo de México (ITAM) on July 1, 2020, under the name "Social and Behavioral Drivers of Individual Compliance with Preventive Measures during the COVID-19 Epidemic in Mexico" (memorandum letter of approval available upon request from the authors.) The questionnaire was pre-tested on a small sample of colleagues and acquaintances, and subject to the IRB's recommendations. Survey respondents were recruited through a Facebook ad campaign and a separate email campaign. The Facebook ad campaign targeted a general audience composed of individuals over 18 years of age living in the Mexican states of Sonora and Guanajuato, it was associated with the official Facebook account of the Inter-American Development Bank (IDB), and it was run by the Knowledge, Innovations and Communications Department of the IDB. The ads can be found in the online supplementary information (S1 Fig). The campaign took place between July 7 and July 21, 2020. The second recruitment channel consisted of an email sent by various secretaries of the Guanajuato state government in Mexico, using, their email distribution lists on Sendy. The list of secretaries that participated in this recruitment process by providing their contact lists are the following: the Secretary of Economic Development, Secretary of Tourism, Secretary of Health and Secretary of Education. This email campaign consisted of two rounds of invitations that took place on July 10 and July 17, 2020 and no exclusion criteria were applied.

The Facebook ads directed respondents to a dedicated project webpage within the IDB website where respondents were able to access the baseline survey. The invitations from the government secretaries did not direct respondents to the dedicated project webpage within the IDB website, instead leading respondents directly to the baseline survey. The baseline survey itself stated on the welcome page that participation was voluntary and that respondents could end the survey at any time and for any reason. It also stated that only those who were at least 18 years of age should respond, even though neither the survey nor the treatments contain any age-inappropriate content. At the end of the survey, we asked respondents whether the individual recommended using her responses in our analysis or not according to how confident the person felt about the quality of the responses. We made clear that there were no consequences if the individual selected "Do not use." A total of 52,507 people clicked on the Facebook ad, yielding 15,542 complete and usable surveys. 14,059 people clicked on the email ad, yielding 7,642 complete and usable surveys. For purposes of the present study, we pooled all usable survey responses from both recruitment channels, for a total of 23,184 respondents.

The first column of Table 2 provides basic descriptive statistics for the control group (these should be close to sample means due to randomization of treatment assignment.) The average respondent is female (66%), completed secondary education (about 58% of the individuals in the sample have completed secondary education or higher), and reported knowing someone who had previously been exposed to COVID-19 (65%), and someone who has died of COVID-19 (58%). About 12% of the sample reported having attended a party in the last 7 days, 43% reported having visited family members in the last 7 days, 74% reported that it is risky to perform activities in enclosed spaces such as gyms or restaurants, and 36% thinks that their neighbors keep social distance from others.

The population in our sample seems to be more female and more educated than the average Mexican person as per the latest available Mexican Population Census. For example, while in our sample 66% of the respondents are female, they are only 51% in the overall population. Moreover, while the share of Mexicans with superior (post-secondary) or university education is about 22%, it is around 50% in our sample. We cannot precisely estimate age in our sample

because respondents were asked to select an age bracket. Our median respondent is in the category [25–39] and the median Mexican person is 29 years old. However, we can estimate that our sample may under-represent older individuals. In Mexico, about 15% of the population is 55 years or older, while it is slightly higher than 10% in our sample (by design, we do not sample minors) (Mexican census and demographic data are available from INEGI at https://www.inegi.org.mx/). As such, our recruitment method may be under-sampling older and less educated individuals who may be less likely to use computers or smartphones, or respond to Facebook ads. In spite of the differences between our sample and the general population, we have no strong reasons to believe that it affects the external validity of the results.

## Experimental design

The experiment consists of a vignette included in the survey depicting a fictional individual, *Mariana*, who has been invited to attend a friend's birthday gathering and must decide whether or not to attend. The vignette is reproduced below. The first paragraph is common to all respondents, while the second paragraph is the experimental prompt. Four different versions of the experimental prompt, and a control condition, were randomized across respondents:

> Mariana lives in Sonora and has been following the public health guidelines related to the current Coronavirus pandemic. A friend invited Mariana and 20 other friends to her birthday party inside her house.

> Mariana knows that her friends think that [*it is*]/[*it is not*] right to attend, [*and*]/[*but*] [*only a few of them*]/[*most of them*] will show up.

The experimental prompts focus on Mariana's reference network (i.e., her friends), as prior research has outlined the importance of one's reference network in shaping one's behavior [22, 51–55]. It is also important to note that our vignette explicitly describes Mariana's "type" as somebody who complies with public health guidance. Making this information explicit could potentially dampen the effect of our treatments (since it provides information on Mariana's unconditional preferences for social distancing), but at the same time it controls for a potential source of unnecessary variation in respondent priors.

Table 1 describes the 2-by-2 experimental design that results from randomizing the empirical and normative expectations prompts. The horizontal dimension varies the content of the empirical expectation (few or most will attend), while the vertical axis that of the normative one (friends consider it appropriate vs. not appropriate to attend). Following Bicchieri et al. [43], our treatment conditions are labeled T1(H/H), T2(H/L), T3(L/H), T4(L/L):

Table 2 describes the balance on covariates measured before the experimental vignette was presented. Judging on the basis of balance on observables, the randomization was successful, as the hypothesis that covariate means are equal across treatment conditions is only rejected twice (p<0.1) out of 39 comparisons.

**Table 1. Treatments and expectations.**

| | | Friends who will attend the party (empirical) | |
| --- | --- | --- | --- |
| | | Few | Most |
| *Friends believe attending the party is appropriate (normative)* | No | **T1 (High/High):** | **T2 (High/Low):** |
| | Yes | **T3 (Low/High):** | **T4 (Low/Low):** |

**Table 2. Balance table.**

| | T1 | Diff w.r.t. T1 (coeff & s.e.) | | | p-value Wald test equality coefficients | | | | Sample Size |
|---|---|---|---|---|---|---|---|---|---|
| | (av & s.e.) | T2 | T3 | T4 | T2 = T3 = T4 | T2 = T3 | T2 = T4 | T3 = T4 | |
| | [1] | [2] | [3] | [4] | [5] | [6] | [7] | [8] | [9] |
| Age (group) | 1.429 | -0.011 | -0.006 | -0.000 | 0.585 | 0.610 | 0.301 | 0.598 | 22,896 |
| | (0.007) | (0.010) | (0.010) | (0.010) | | | | | |
| 1.Female | 0.660 | 0.003 | 0.004 | -0.001 | 0.799 | 0.835 | 0.655 | 0.511 | 23,184 |
| | (0.006) | (0.009) | (0.009) | (0.009) | | | | | |
| Education (group) | 2.580 | 0.021* | 0.008 | 0.018 | 0.548 | 0.293 | 0.803 | 0.427 | 22,925 |
| | (0.009) | (0.012) | (0.012) | (0.012) | | | | | |
| 1.Exposed Covid | 0.649 | 0.006 | 0.008 | 0.003 | 0.805 | 0.758 | 0.726 | 0.510 | 22,625 |
| | (0.006) | (0.009) | (0.009) | (0.009) | | | | | |
| 1.Death Covid | 0.576 | -0.000 | -0.000 | 0.002 | 0.958 | 0.994 | 0.803 | 0.796 | 23,184 |
| | (0.006) | (0.009) | (0.009) | (0.009) | | | | | |
| 1.Older 65 | 0.265 | -0.003 | -0.002 | -0.004 | 0.960 | 0.917 | 0.859 | 0.777 | 23,093 |
| | (0.006) | (0.008) | (0.008) | (0.008) | | | | | |
| 1.Exposed H1N1 | 0.186 | 0.010 | 0.008 | 0.012* | 0.832 | 0.730 | 0.796 | 0.546 | 23,184 |
| | (0.005) | (0.007) | (0.007) | (0.007) | | | | | |
| Prob Infection | 51.344 | -0.062 | 0.206 | 0.098 | 0.879 | 0.613 | 0.765 | 0.839 | 22,964 |
| | (0.375) | (0.532) | (0.528) | (0.534) | | | | | |
| Prob Hospital | 45.429 | 0.080 | -0.317 | -0.320 | 0.621 | 0.397 | 0.398 | 0.993 | 22,988 |
| | (0.336) | (0.474) | (0.470) | (0.476) | | | | | |
| 1.Attend Party | 0.125 | -0.006 | -0.001 | -0.002 | 0.701 | 0.429 | 0.521 | 0.885 | 23,087 |
| | (0.004) | (0.006) | (0.006) | (0.006) | | | | | |
| 1.Visit | 0.428 | -0.008 | 0.005 | -0.014 | 0.116 | 0.183 | 0.478 | 0.0411 | 23,085 |
| | (0.007) | (0.009) | (0.009) | (0.009) | | | | | |
| 1.Risky Inside | 0.734 | 0.005 | -0.002 | 0.002 | 0.665 | 0.367 | 0.672 | 0.635 | 23,184 |
| | (0.006) | (0.008) | (0.008) | (0.008) | | | | | |
| 1.Social Distance | 0.360 | 0.008 | -0.008 | -0.004 | 0.189 | 0.080 | 0.184 | 0.681 | 23,098 |
| | (0.006) | (0.009) | (0.009) | (0.009) | | | | | |

*Notes*: Each row shows statistics for a different observable variable we have. Column [1] shows the sample average and the standard deviation in parenthesis for the control group -in this case, individuals in T1. Columns [2]-[4] shows the regression coefficient and the standard error in parenthesis corresponding to an OLS regression -observable is the dependent variable and the treatment variables are the independent ones. Standard errors are robust.

*** $p<0.01$,

** $p<0.05$,

* $p<0.1$.

Columns [5]-[8] shows the p-value of a test of equality of coefficients. Column [9] shows the sample size for each regression. Variables *Age* and *Education* are tabulated according to ranges; as such they are categorical, with a higher category number referring to an older age and more years of education, respectively. 1.x refers to dummy variables.

Our outcome variables come from two questions immediately following exposure to the vignette: i) whether the respondent thinks that Mariana will or will not attend to the gathering, and ii) whether the respondent approves or does not approve of Mariana attending the gathering. Following the literature [16, 41, 43], our main hypothesis is:

**H1**: Those exposed to high empirical and normative expectations (T1) will be more likely to predict that Mariana will social distance and refrain from attending the gathering than respondents exposed to the low empirical and normative expectations (T4).

Ex-ante, we remain agnostic about the relative effects of the "incongruent" sets of expectations in treatments T2 (high empirical expectations and low normative expectations) and T3 (low empirical expectations and high normative expectations), as do Bicchieri et al. [43].

### Estimation strategy

We estimate the following linear probability model on the outcome data:

$$y_i = \alpha + \beta T_{2-4} + \lambda X_i + u_i, \tag{1}$$

where $y_i$ is the value of a dependent variable for respondent $i$ (0 = will not / should not attend, 1 = will / should attend), and $T_{2-4}$ is an indicator variable taking the value of 1 when $i$ was assigned to any of treatment branches 2, 3, or 4, with $T_1$ as the reference category. The coefficient $\beta$ represents the difference in the mean value of the dependent variable between those assigned to treatments 2, 3, or 4, on the one hand, and those assigned to Treatment 1. $X$ is a vector of controls. It includes all observable characteristics available from the survey: age, female, education, exposed to COVID, Death due to COVID, Older than 65 living at home, had H1N1 in the past, perception about the probability of infection, and the probability of ending up in the hospital, whether the individual or a family member went to a party or visited family in the last 7 days, their perception about how risky it is to be inside, and their evaluating regarding how well neighbors comply with social distancing guidelines. We additionally estimate specifications with separate indicator variables for each of the treatment conditions:

$$y_i = \alpha + \beta_2 T_2 + \beta_3 T_3 + \beta_4 T_4 + \lambda X_i + v_i, \tag{2}$$

where $T_j$ are indicator variables for treatment assignment to treatments $j$ = 2, 3, 4. In this case, the coefficients $\beta_j$ estimate average treatment effects of Treatment $j$ in comparison with the reference Treatment 1. The main coefficient of interest is $\beta_4$, which measures the difference between the scenario where Mariana expects few friends to attend the gathering and few to approve of attending (T1) versus one where Mariana expects many to attend and many to approve of attending (T4). $X$ is a vector of controls, as already described. Both equations estimate intent-to-treat effects.

## Results and discussion

### Predicted attendance

Columns 1-4 of Table 3 display the results for the dependent variable concerning respondents' predictions about whether Mariana *will or will not* attend the gathering. The first column presents estimates of Eq 1 without control variables. Respondents assigned to scenarios T2, T3, or T4 on average expected that Mariana would be about 7 percentage points (p<.01) more likely to attend the gathering than those assigned to T1, the scenario where Mariana expected few friends to attend and few friends to approve of attending. This is a large effect, equivalent to 28% of the predicted probability that Mariana would attend in the reference category T1. The estimated $\beta$ is very similar—in fact slightly larger—when adding a battery of individual-level controls (column 2), state fixed effects (column 3), or municipality fixed effects (column 4).

Fig 1 displays the respective marginal effects of the joint treatment variable and the control variables. The panel on the left corresponds to the specification in column 2 of Table 3. The probability of responding that Mariana will attend the party decreases with respondent age (3 pp per age category) and it is lower for female respondents (4 pp). As one might expect, the prediction is also lower for respondents who believe the risk of indoor contagion is high (4 pp), and for those who report that their neighbors practice social distancing (4 pp). On the

**Table 3. Treatment effects.**

|  | Mariana will attend | | | | Mariana should attend | | | |
|---|---|---|---|---|---|---|---|---|
|  | (1) | (2) | (3) | (4) | (5) | (6) | (7) | (8) |
| T (T2+T3+T4) | 0.073*** | 0.076*** | 0.076*** | 0.077*** | 0.001 | 0.001 | 0.002 | 0.002 |
|  | (0.007) | (0.007) | (0.007) | (0.007) | (0.003) | (0.003) | (0.003) | (0.003) |
| Constant | 0.264*** | 0.321*** | 0.381*** | 0.339*** | 0.033*** | 0.107*** | 0.142*** | 0.129*** |
|  | (0.006) | (0.020) | (0.066) | (0.041) | (0.002) | (0.009) | (0.033) | (0.020) |
| T2 | 0.098*** | 0.100*** | 0.100*** | 0.101*** | -0.002 | -0.000 | -0.000 | 0.001 |
|  | (0.009) | (0.009) | (0.009) | (0.009) | (0.003) | (0.003) | (0.003) | (0.003) |
| T3 | 0.055*** | 0.058*** | 0.058*** | 0.059*** | 0.004 | 0.004 | 0.004 | 0.005 |
|  | (0.009) | (0.009) | (0.009) | (0.009) | (0.003) | (0.003) | (0.003) | (0.003) |
| T4 | 0.067*** | 0.069*** | 0.070*** | 0.071*** | -0.000 | 0.000 | 0.000 | 0.001 |
|  | (0.009) | (0.009) | (0.009) | (0.009) | (0.003) | (0.003) | (0.003) | (0.003) |
| Constant | 0.264*** | 0.322*** | 0.378*** | 0.338*** | 0.033*** | 0.107*** | 0.143*** | 0.129*** |
|  | (0.006) | (0.020) | (0.066) | (0.041) | (0.002) | (0.009) | (0.033) | (0.020) |
| Observations | 21,882 | 20,511 | 20,511 | 20,511 | 22,744 | 21,264 | 21,264 | 21,264 |
| Controls | No | Yes | Yes | Yes | No | Yes | Yes | Yes |
| Fixed Effects | No | No | State | Municipality | No | No | State | Municipality |
| T2 = T3 = T4 | 0.000 | 0.000 | 0.000 | 0.000 | 0.189 | 0.412 | 0.394 | 0.437 |
| T2 = T3 | 0.000 | 0.000 | 0.000 | 0.000 | 0.078 | 0.229 | 0.208 | 0.220 |
| T2 = T4 | 0.001 | 0.001 | 0.001 | 0.001 | 0.675 | 0.935 | 0.887 | 0.808 |
| T3 = T4 | 0.192 | 0.228 | 0.202 | 0.198 | 0.180 | 0.263 | 0.266 | 0.328 |

*Notes*: The first block shows the results for the joint treatments. The second block for each treatment individually. Each row shows the regression coefficients and the standard error in parenthesis corresponding to an OLS regression. Dependent variables take the value 0-1. Standard errors are robust.

*** $p<0.01$,

** $p<0.05$,

* $p<0.1$.

Controls include: sex, age, education, exposed to Covid, death to Covid, older than 65 at home, knows infected H1N1, belief about infection probability, belief about hospitalization probability, attends party, visits family, risk inside evaluation, and others practice social distancing.

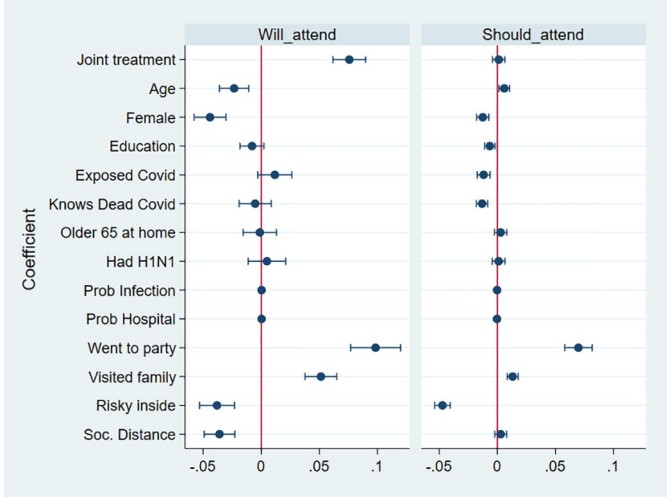

**Fig 1. Treatment effects—joint treatment and controls.** This figure shows the coefficients for the joint treatment variable and the coefficients for the control variables. It corresponds to columns [2] and [6] in Table 3.

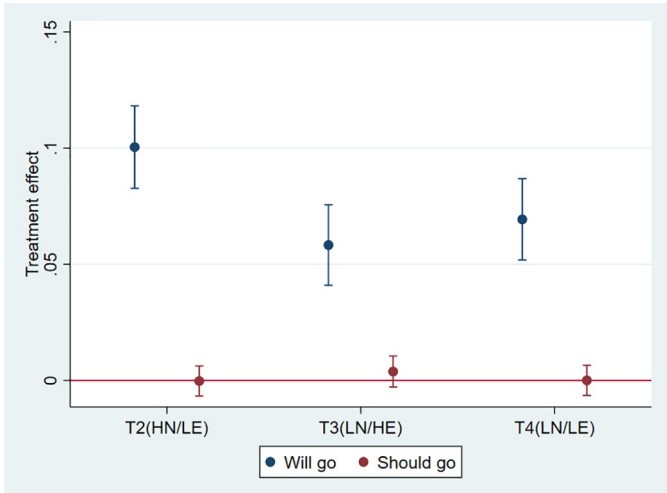

**Fig 2. Treatment effects.** This figure shows the treatment effects for the two dependent variables. They correspond to columns [2] and [6] in Table 3.

contrary, the predicted probability that Mariana will attend increases for respondents who report having attended a party themselves in the last week (10 pp), and for those who report having visited friends or family recently (5 pp).

The lower part of the Table 3 shows estimates from Eq 2. The key coefficient is $\beta_4$, as it represents a test of hypothesis *H*1. The estimated value of $\beta_4$ is about 7 percentage points (P<.01), implying that assignment to the low normative and low empirical expectation vignette (T4) increases the predicted likelihood of answering that Mariana will attend the gathering, in comparison with T1 (the high normative and empirical expectations treatment), by about 25%. This effect is very large and is consistent with hypothesis *H*1, that those exposed to low empirical and normative expectations (T4) will be more likely to predict that Mariana will not social distance compared to those who are exposed to high empirical and normative expectations (T1). Fig 2 shows the coefficients in graphical terms. Table 3, last four rows, shows the p-value of a test of equality of coefficients (Wald test) for evaluating the differences between T1 and T4, and the "incongruent" treatments [43]. Results show that the coefficient for T2, $\beta_2$ is higher and is statistically different than those for T3 and T4 $\beta_3$ and $\beta_4$, (p<0.01). $\beta_3$ is lower but not statistically different than $\beta_4$. We discuss the implications of these results in the next section.

## Respondent approval of attending

Columns (5)-(8) in Table 3 display estimates for our second dependent variable: respondent views on whether Mariana *should or should not attend* the party. In models 1 and 2, and in all specifications, we find that the effect is a precisely estimated zero. Treatment arms are not statistically different from each other either. While we can only speculate about the reason behind this result, one possibility is that it reflects a ceiling effect: almost every respondent, regardless of treatment assignment, expressed the view that Mariana should not attend. This is consistent with the universal approval of preventive guidelines documented in surveys of the Mexican public. It also suggests that there is a disconnect between such approval an actual behavior, or between approval and the predicted behavior of others. Clearly, however, our results lend no

support to the possibility that the effects we find on predicted behavior are mediated by effects on normative views about such behavior.

## Conclusion

Even as COVID-19 infection rates are again on the rise in many countries, lockdown fatigue has set in and opposition to social distancing measures is stronger than ever. Voluntary compliance, therefore, is of paramount importance. Our results suggest that policies that harness social norms to that end could be of help.

Specifically, our study shows that predicted compliance with social norms is greatest when the fictional character in the vignette, Mariana, i) expects few of her friends to attend, and ii) believes few of her friends would approve of her attending. Whenever either of these conditions fails to hold (or both do), predicted attendance rises significantly. In other words, both high empirical and high normative expectations appear to be necessary to increase compliance with social distancing. This suggests that norms-based information campaigns can be more effective by targeting both kinds of expectations. It also suggests that undermining compliance is easier than sustaining it, as reducing either empirical or normative expectations suffices—in our study—to discourage social distancing.

Our results provide mixed support for various ideas in the literature on the relative importance of normative versus empirical expectations. On the one hand, comparing the effects of treatment branches T2 (high empirical, low normative) versus T3 (low empirical, high normative) suggests that empirical expectations matter more than normative expectations, as claimed in [56]. At the same time, the estimated effect of treatment T4 (high empirical, high normative) is smaller in magnitude than, and statistically different from, that of treatment T2. This is surprising, since one might expect that when normative and empirical expectations are aligned (T4), the effect on behavior should be larger—yet this is not what we find. We take our results on the mixed treatments (T2 and T3) as an indication that empirical and normative expectations may interact in ways that are poorly understood (perhaps some form of crowding out is at work) and merit further research.

Our study design, of course, has limitations. First, it is not obvious that the intensity of treatment is comparable across arms: it could be that changes in the perceived empirical expectations are greater than a change in normative expectations. Second, our results ought to be interpreted in the context of the fact that Mariana is said, in the vignette, to generally comply with public health guidelines. Therefore, respondents may infer that Mariana may care more about what her friends like her do (T1 and T2) than those friends who do not think like her (T3 and T4). Lastly, our estimations are based on the perception of participants on how others (Mariana) would behave in this scenario. We, therefore, cannot assure that participants would act similarly if they found themselves in a similar position.

Our findings contribute to the general research on the relationship of social norms with behavior and are relevant for the design of communication strategies in both the public and private sectors. Highlighting that others are not complying is likely to reduce compliance, and this could be an unintended byproduct of news coverage about noncompliance. Politicization of the guidelines, and active and public repudiations of norms, can also lead to further erosion of compliance. Additionally, targeting normative expectations—what people ought to be doing—will likely not suffice to induce the desired behaviors unless people also expect others to comply. Thus, information highlighting others' compliance and targeting normative expectations at the same time are likely to play an essential role in any successful information campaign seeking to encourage individuals to adopt preventive behaviors.

## Supporting information

**S1 Fig. Facebook ads—recruitment.** The figure shows an example of the ads used for recruitment.
(PDF)

**S1 Appendix. Survey questions.** This document presents all the questions (in Spanish—original language of the survey—and English) used to construct dependent and treatment variables, as well as covariates.
(PDF)

**S1 File. Data set.** This .dta file contains the underlying data set used to reach the conclusions drawn in this paper.
(DTA)

**S2 File. Regressions code.** This .do file allows readers to replicate the results of the paper using the S1 File.
(DO)

## Acknowledgments

We are thankful to the editor, two anonymous reviewers, Martín Ardanaz, Ivanna Valverde, Fernando Cafferata, Ana María Rojas Méndez, Simeon Schachtele, Jorge Streb, Eugen Dimant and participants in the Impact Evaluation Meeting on COVID-19 for their comments. All remaining errors are our own.

## Author Contributions

**Conceptualization:** Déborah Martínez, Cristina Parilli, Carlos Scartascini, Alberto Simpser.

**Data curation:** Carlos Scartascini.

**Formal analysis:** Carlos Scartascini, Alberto Simpser.

**Investigation:** Carlos Scartascini.

**Methodology:** Carlos Scartascini.

**Project administration:** Déborah Martínez, Cristina Parilli, Alberto Simpser.

**Supervision:** Déborah Martínez.

**Writing – original draft:** Cristina Parilli, Carlos Scartascini, Alberto Simpser.

**Writing – review & editing:** Cristina Parilli, Carlos Scartascini, Alberto Simpser.

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
