## [Decision Letter · Decision Letter 0]

5 Jan 2021

PONE-D-20-35018

Let's (not) get together! The role of social norms on social distancing during COVID-19

PLOS ONE

Dear Dr. Scartascini,

Thank you for submitting your manuscript to PLOS ONE. After careful consideration, we feel that it has merit but does not fully meet PLOS ONE’s publication criteria as it currently stands. Therefore, we invite you to submit a revised version of the manuscript that addresses the points raised during the review process.

Please find below the reviewers' comments, as well as those of mine.

We look forward to receiving your revised manuscript.

Kind regards,

Valerio Capraro

Academic Editor

PLOS ONE

Journal Requirements:

2.Thank you for including your ethics statement:

"Data collection was approved by ITAM's IRB and performed under the ethics guidelines of the IDB"

3.Please provide additional details regarding participant consent. In the ethics statement in the Methods and online submission information, please ensure that you have specified (1) whether consent was informed and (2) what type you obtained (for instance, written or verbal, and if verbal, how it was documented and witnessed). If your study included minors, state whether you obtained consent from parents or guardians. If the need for consent was waived by the ethics committee, please include this information.

4. In your Methods section, please provide additional information about the participant recruitment method and the demographic details of your participants. Please ensure you have provided sufficient details to replicate the analyses such as: a) the recruitment date range (month and year), b) a description of any inclusion/exclusion criteria that were applied to participant recruitment, c) a table of relevant demographic details, d) a statement as to whether your sample can be considered representative of a larger population, e) a description of how participants were recruited, and f) descriptions of where participants were recruited and where the research took place.

5. Please include additional information regarding the survey or questionnaire used in the study and ensure that you have provided sufficient details that others could replicate the analyses. For instance, if you developed a questionnaire as part of this study and it is not under a copyright more restrictive than CC-BY, please include a copy, in both the original language and English, as Supporting Information. Moreover, please include more details on how the questionnaire was pre-tested, and whether it was validated.

6.In your Data Availability statement, you have not specified where the minimal data set underlying the results described in your manuscript can be found. PLOS defines a study's minimal data set as the underlying data used to reach the conclusions drawn in the manuscript and any additional data required to replicate the reported study findings in their entirety. All PLOS journals require that the minimal data set be made fully available. For more information about our data policy, please see http://journals.plos.org/plosone/s/data-availability.

7. Your ethics statement should only appear in the Methods section of your manuscript. If your ethics statement is written in any section besides the Methods, please move it to the Methods section and delete it from any other section. Please ensure that your ethics statement is included in your manuscript, as the ethics statement entered into the online submission form will not be published alongside your manuscript.

8.We note that Figure(s) A3 in your submission contain copyrighted images. All PLOS content is published under the Creative Commons Attribution License (CC BY 4.0), which means that the manuscript, images, and Supporting Information files will be freely available online, and any third party is permitted to access, download, copy, distribute, and use these materials in any way, even commercially, with proper attribution. For more information, see our copyright guidelines: http://journals.plos.org/plosone/s/licenses-and-copyright.

a.    You may seek permission from the original copyright holder of Figure(s) A3 to publish the content specifically under the CC BY 4.0 license.

9. Please ensure that you refer to Figure 2 in your text as, if accepted, production will need this reference to link the reader to the figure.

10.Thank you for stating the following in the Financial Disclosure section:

"The author(s) received no specific funding for this work"

We note that one or more of the authors are employed by a commercial company: Inter-American Development Bank

Additional Editor Comments (if provided):

I have now collected two reviews from two experts in the field. The reviews are somehow split, one recommends rejection and the other recommends minor revision. The reason underlying the negative review is that the paper is at best incremental. However, the results are sound. Since novelty is not a key requirement to get published in Plos One, I have then decided to follow the second reviewer and invite you to revise the paper. In the revision, please try to address also the comments from the negative reviewer: it will not hurt. Also, I would like to add a couple more comments, mainly related to the literature review. (i) I think that the "perspective article" on what social and behavioural science can do to support pandemic response, published by Van Bavel et al. in Nature Human Behaviour, can be a useful general introductory references. (ii) the literature review regarding which messages affect pandemic response is a bit incomplete. Here's a list of such papers that I know in the topic, including mine: Capraro & Barcelo (2020a); Capraro & Barcelo (2020b); Everett et al. (2020), Heffner et al. (2020), Jordan et al. (2020), Lunn et al (2020), Pfattheicher et al. (2020). It's possible that there are others one. Please include a detailed literature review and discuss how your paper fits in this literature.

I am looking forward for the revision.

References

Capraro, V., & Barcelo, H. (2020a). The effect of messaging and gender on intentions to wear a face covering to slow down COVID-19 transmission. Journal of Behavioral Economics for Policy.

Capraro, V., & Barcelo, H. (2020b). Priming reasoning increases intentions to wear a face covering to slow down COVID-19 transmission. arXiv preprint arXiv:2006.11273.

Everett, J. A., Colombatto, C., Chituc, V., Brady, W. J., & Crockett, M. (2020). The effectiveness of moral messages on public health behavioral intentions during the COVID-19 pandemic. https://psyarxiv.com/9yqs8/

Heffner, J., Vives, M. L., & FeldmanHall, O. (2020). Emotional responses to prosocial messages increase willingness to self-isolate during the COVID-19 pandemic. Personality and Individual Differences, 170, 110420.

Jordan, J., Yoeli, E., & Rand, D. (2020). Don’t get it or don’t spread it? Comparing self-interested versus prosocially framed COVID-19 prevention messaging. https://psyarxiv.com/yuq7x

Lunn, P. D., Timmons, S., Barjaková, M., Belton, C. A., Julienne, H., & Lavin, C. (2020). Motivating social distancing during the Covid-19 pandemic: An online experiment. Social Science & Medicine, 113478.

Pfattheicher, S., Nockur, L., Böhm, R., Sassenrath, C., & Petersen, M. B. (In press). The emotional path to action: Empathy promotes physical distancing during the COVID-19 pandemic. Psychological Science.

Van Bavel, J. J., et al. (2020). Using social and behavioural science to support COVID-19 pandemic response. Nature Human Behaviour, 4, 460-471.

Reviewers' comments:

Reviewer's Responses to Questions

**Comments to the Author**

1. Is the manuscript technically sound, and do the data support the conclusions?

Reviewer #1: No

Reviewer #2: Partly

2. Has the statistical analysis been performed appropriately and rigorously? 

Reviewer #1: No

Reviewer #2: Yes

3. Have the authors made all data underlying the findings in their manuscript fully available?

Reviewer #1: No

Reviewer #2: Yes

4. Is the manuscript presented in an intelligible fashion and written in standard English?

Reviewer #1: No

Reviewer #2: Yes

5. Review Comments to the Author

Reviewer #1: This is a quick review. I liked reading the paper. There is a timely implication for COVID adherence. My concerns are:

> Several endogenous factors such as information levels, information sources, person's knowledge about COVID, social responsibilities/connections are not controlled for.

> Answers to survey questions are quite perceptual--than realistic, that poses validity aspects of the study

> The contributions and implications, at its best, can be incremental to the existing research in this area.

Reviewer #2: Ms. Ref. No.: PONE-D-20-35018

Title: "Let's (not) get together! The role of social norms on social distancing during COVID-19

PLOS ONE

Overall, this is a well-written study that applies social norms to the current pandemic context. The authors provide a thorough discussion of the current work in these areas as well as topics that should be considered in future research. The authors provide a clear overview of previous work in these areas and the necessity of the current work. The paper is organized well and written with very clear, easy-to-follow language that will make this review useful for a wide audience of scholars in the fields of social psychology, communication, and beyond. I believe this is a timely contribution to the literature and should be accepted with several minor revisions.

(1) The authors list that “the average respondent is female, completed secondary education, and reporting knowing somebody who had previously been exposed to COVID-19.” It would be much more beneficial to the reader to have the actual demographic information, rather than this general statement. The authors could include the specific percent of the sample that is female, the specific percent that knew someone who had COVID-19, and the educational breakdown. I have a similar comment about the disclosure of age- please report the M and SD of age, rather than the general statement that the sample is “older… than the average Mexican person” (pg 5).

(2) Page 8 refers to a “Figure ??” without guiding the reader to the correct figure. Please correct so we know what visual to refer to.

(3) The results section is unclear in regards to the differences between T1, T2, T3, and T4. A specific hypothesis was made that compares T1 and T4, and no specific hypotheses are made about the potential differences between the other conditions. However, the researchers should still probe these differences and explain them more thoroughly. It would be beneficial for the authors to run ANOVAs to compare the DVs by each condition and to report these results. It is not totally clear how these 4 conditions are similar or different based on the brief reporting on pg 8. A greater explanation of the results section overall would also help with clarifying the findings.

(4) Similarly, the discussion on page 9 states that both empirical and normative expectations are important to increase compliance. Yet, it is not clear if the data suggested that- if so, then T1, T2, and T3 should all function the same; only T4 should show increased compliance if both of these norms are necessary for individuals to comply as the authors state on page 9. Again, reporting the results of an ANOVA will help demonstrate to the reader the strength of the authors’ claims.

(5) Small correction: some of the citations in the manuscript still follow old APA guidelines and should be updated to follow the new standards (e.g. listing work in author alphabetical order instead of by date of publication; using an ampersand instead of writing out “and” between author names)

(6) It would be helpful to submit the appendix materials in English as well, so English-speaking readers can understand the materials participants were exposed to.

I enjoyed reading this manuscript, and am confident the authors can address my minor suggestions to help strengthen what is already a great paper.

6. PLOS authors have the option to publish the peer review history of their article (what does this mean?). If published, this will include your full peer review and any attached files.

Reviewer #1: No

Reviewer #2: No

---

## [Author Response · Author response to Decision Letter 0]

5 Feb 2021

Reviewers' comments:

Reviewer #1: 

> Several endogenous factors such as information levels, information sources, person's knowledge about COVID, social responsibilities/connections are not controlled for.

Because we are comparing responses of individuals who were randomly assigned to different vignettes we don’t expect that our results would be influenced by third variables -we have no reason to believe that any of these would distribute in a non-random fashion, particularly given that in Table 2 we show that treatments are balanced for all the pre-determined observable covariates we have available. Still, in the regressions we control for: age, female, education, whether the respondent or their family have been exposed to COVID, whether the respondent knows somebody who died due to COVID, whether there is somebody 65 or older living at home with the respondent, whether the respondent had H1N1 in the past, the respondent’s perception about the probability that they will become infected with COVID, the respondent’s perceived probability that they will end up in the hospital, whether the respondent or a family member attended a party or visited family in the last 7 days, the respondent’s perception about the degree of risk of contagion associated with spending time indoors with other people, and the respondent’s evaluation regarding the degree to which their neighbors comply with social distancing guidelines.

> Answers to survey questions are quite perceptual--than realistic, that poses validity aspects of the study

Our approach---perhaps like any other approach---has advantages and disadvantages. The reviewer is right in asserting that survey experiments could reflect intentions rather than actual behavior (stating that “I will attend a party” does not guarantee that the respondent would in fact attending a party). Moreover, the responses we are eliciting are about the respondent’s expectations about the behavior of a third party (“Mariana”), not about the respondent’s own behavior. Another disadvantage---associated with asking respondents about a hypothetical situation---is that they could fail to pay attention or they could misunderstand the situation or the questions we asked (which would perhaps be less likely to happen if we were asking about their own behavior). We tried to minimize this problem by making the vignette and the questions as simple as possible and by presenting individuals with a very familiar scenario: a friend inviting Mariana to attend a birthday party. In our pretesting there were no instances of misunderstanding. One advantage of our approach is that asking about a third party could potentially mitigate social desirability bias, as argued by a large body of research.

Still, even if responses were biased, so long as biases were similar across all treatment branches our estimates of treatment effects would remain unbiased. The reason is that our estimates consist of differences in average responses across two treatment branches. It is possible for differences to be unbiased even if levels are biased (one remaining danger when levels are biased is that they could create a ceiling effect for the differences across treatment branches---but that is not the case in our analysis). Because our randomization resulted in well-balanced treatment branches, it is reasonable to suppose that social desirability bias (and other biases) are also likely to be balanced across treatment branches.

Reviewer #2

(1) The authors list that “the average respondent is female, completed secondary education, and reporting knowing somebody who had previously been exposed to COVID-19.” It would be much more beneficial to the reader to have the actual demographic information, rather than this general statement. The authors could include the specific percent of the sample that is female, the specific percent that knew someone who had COVID-19, and the educational breakdown. I have a similar comment about the disclosure of age- please report the M and SD of age, rather than the general statement that the sample is “older… than the average Mexican person” (pg 5).

We appreciate the reviewer’s suggestion. All the demographic characteristics are now included in Table 1. Unfortunately, in some of the cases, the information cannot be directly benchmarked with population data because some of the questions in our survey were asked in terms of brackets (for example, age). Still, we have improved the text substantially by comparing the data we have with the Mexican census data. The text now reads:

“For example, 51% of Mexicans female but 66% are female in our sample, and the share of Mexicans with university education is 22% but it is 50% in our sample. The average Mexican is 29 years old, the median age range in our sample is 25 to 39 years of age. Older individuals, however, appear to be underrepresented in our sample: 15% of Mexicans are 55 or older, while only a bit over 10% of respondents in our sample are. Recall also that our sample excludes minors. (Mexican census and demographic data are available from INEGI at https://www.inegi.org.mx/)”

(2) Page 8 refers to a “Figure ??” without guiding the reader to the correct figure. Please correct so we know what visual to refer to.

We are sorry for this typo. We have now corrected it in the text and made sure there are no other similar typos in the text

(3) The results section is unclear in regards to the differences between T1, T2, T3, and T4. A specific hypothesis was made that compares T1 and T4, and no specific hypotheses are made about the potential differences between the other conditions. However, the researchers should still probe these differences and explain them more thoroughly. It would be beneficial for the authors to run ANOVAs to compare the DVs by each condition and to report these results. It is not totally clear how these 4 conditions are similar or different based on the brief reporting on pg 8. A greater explanation of the results section overall would also help with clarifying the findings.

We agree with the reviewer that in an effort to keep the paper short, we were vague in our description of the ‘Results’ section -we discussed differences more clearly instead in the ‘Discussion’ section. Now, we make sure to describe the differences between T1, T2, T3, and T4 explicitly, and then continue the discussion of the implications of these results in the discussion section. 

In response to the reviewer’s suggestion to run ANOVAs, we have performed some additional analysis in line with the requests by the reviewer (shown further below in this document). The first table shows a contingency analysis table. It compares the proportion of people who responded that Mariana will/should attend the party across all treatment assignment categories. As shown in the Table, and in line with our regression table in the paper, T2, T3, and T4 are higher than T1 for the first dependent variable. According to the Chi square statistic they are jointly statistically different. 

We also perform ANOVA estimations in two forms: (i) looking at the joint significance of the differences in the outcome variable (Mariana will/should attend the gathering) among all treatment assignment categories -first column; (ii) estimations by pairs of treatment categories (columns 2 to 5). Results are basically the same, showing significant differences across treatments.

We also present a Kruskal-Wallis non-parametric test, given that the nature of the dependent variable might reject the normality assumption over the errors that the ANOVA analysis presupposes. 

Regardless of the method adopted, the results are exactly the same. For the question about whether Mariana will attend the party: coefficients are different for T2, T3, and T4 when compared to T1, and they are also different between each other. In contrast, all coefficients are statiscally the same for the question about whether Mariana should attend the party regardless of the method used.

Because the results are consistent across the board, and the Table in the manuscript already includes t-tests for the equality of coefficients, for the sake of space and conciseness we opted for not including the additional results presented here in the paper. But we remain open to doing so if the reviewer believes that the paper would benefit if we did.

(4) Similarly, the discussion on page 9 states that both empirical and normative expectations are important to increase compliance. Yet, it is not clear if the data suggested that- if so, then T1, T2, and T3 should all function the same; only T4 should show increased compliance if both of these norms are necessary for individuals to comply as the authors state on page 9. Again, reporting the results of an ANOVA will help demonstrate to the reader the strength of the authors’ claims.

We have now clarified the results further. As in Bichieri et al (2020), we can compare clearly T1 and T4, and we have no strong priors about T2 and T3. Still, the fact that T2 is higher and statistically different than T3 and T4 is surprising. As we indicate in the paper, comparing the effects of treatment branches T2 vs. T3 suggests that empirical expectations matter more than normative expectations, as claimed in Bicchieri and Xiao (2009). At the same time, the finding that T4 < T2 is surprising, since one might expect that when normative and empirical expectations are aligned (T4), the effect on behavior should be larger---yet this is not what we find. Therefore, our results provide mixed support for various ideas in the literature about the relative importance of normative versus empirical expectations. We take our results on the mixed treatments (T2 and T3) as an indication that empirical and normative expectations may interact in ways that are poorly understood (perhaps some form of crowding out is at work) and merit further research. 

Our study design, of course, has limitations. First, it is not obvious that the intensity of treatment is comparable across arms. In particular, it is conceivable that the variation across treatment branches in empirical expectations is not commensurable with the variation across treatment branches in normative expectations---these are two different dimensions and they may operate on different subjective scales. Second, our results ought to be interpreted in the context of the fact that Mariana is described, in the vignette, as somebody who generally complies with public health guidelines. Therefore, respondents could infer that Mariana cares more about what her friends who are similar to her do (T1 and T2) than about what friends who do not think like her do (T3 and T4). Lastly, our estimations are based on the perception of respondents about how a third party (that is, Mariana) would behave in the given scenario. We cannot guarantee that respondents themselves would act similarly if they were put in a similar position.

(5) Small correction: some of the citations in the manuscript still follow old APA guidelines and should be updated to follow the new standards (e.g. listing work in author alphabetical order instead of by date of publication; using an ampersand instead of writing out “and” between author names)

We now follow the template for PLOS journals and bibliography recommendations in the submission guidelines.

(6) It would be helpful to submit the appendix materials in English as well, so English-speaking readers can understand the materials participants were exposed to.

We appreciate this comment. We have translated all the material in the appendix to English.

---

## [Editor Report · Decision Letter 1]

8 Feb 2021

Let's (not) get together! The role of social norms on social distancing during COVID-19

PONE-D-20-35018R1

Dear Dr. Scartascini,

We’re pleased to inform you that your manuscript has been judged scientifically suitable for publication and will be formally accepted for publication once it meets all outstanding technical requirements.

Kind regards,

Valerio Capraro

Academic Editor

PLOS ONE
---

## [Editor Report · Acceptance letter]

11 Feb 2021

PONE-D-20-35018R1 

Let’s (not) get together! The role of social norms on social distancing during COVID-19  

Dear Dr. Scartascini:

I'm pleased to inform you that your manuscript has been deemed suitable for publication in PLOS ONE. Congratulations! Your manuscript is now with our production department. 

Kind regards, 

on behalf of

Dr. Valerio Capraro 

Academic Editor

PLOS ONE